# RegionMed-CLIP: A Region-Aware Multimodal Contrastive Learning Pre-trained Model for Medical Image Understanding

## Abstract

Medical image understanding plays a crucial role in enabling automated diagnosis and data-driven clinical decision support. However, its progress is impeded by two primary challenges: the limited availability of high-quality annotated medical data and an overreliance on global image features, which often miss subtle but clinically significant pathological regions. To address these issues, we introduce RegionMed-CLIP, a region-aware multimodal contrastive learning framework that explicitly incorporates localized pathological signals along with holistic semantic representations. The core of our method is an innovative region-of-interest (ROI) processor that adaptively integrates fine-grained regional features with the global context, supported by a progressive training strategy that enhances hierarchical multimodal alignment. To enable large-scale region-level representation learning, we construct MedRegion-500k, a comprehensive medical image-text corpus that features extensive regional annotations and multilevel clinical descriptions with rigorous quality control through human-AI collaborative annotation. Extensive experiments on image-text retrieval, zero-shot classification, and visual question answering tasks demonstrate that RegionMed-CLIP achieves significant improvements over state-of-the-art vision language models. Our results highlight the critical importance of region-aware contrastive pre-training and position RegionMed-CLIP as a foundation for advancing multimodal medical image understanding.

## 1 Introduction

Understanding medical image serves as a cornerstone of modern healthcare, enabling both automated disease detection and evidence-based clinical decision making Esteva et al. (2021). In recent years, significant advances have been made with vision language pre-training models such as CLIP Radford et al. (2021), which have been adapted to the medical domain in models like MedCLIP Wang et al. (2022) and BiomedCLIP Huang et al. (2023). However, several critical challenges continue to impede the broader applicability of these models in medical imaging. First, the creation of high-quality annotated medical datasets remains a significant barrier, as it requires expert knowledge and is subject to stringent privacy regulations Irvin et al. (2019); Johnson et al. (2019). Additionally, current multimodal models typically focus on aligning global image features with textual descriptions, often at the expense of fine-grained pathological details that are crucial for accurate clinical diagnosis Lu et al. (2022); Zhou et al. (2023). These limitations can lead to suboptimal model performance in real-world diagnostic tasks where the ability to recognize subtle, localized abnormalities is essential.

In addition to these challenges, the lack of sufficient region-level annotations continues to restrict a model's ability to learn spatially specific pathology. To address these gaps, we introduce the MedRegion-500k dataset, a comprehensive resource designed for fine-grained multimodal learning across a wide range of clinical scenarios. Although MedRegion-500k contains approximately 500,000 image–text pairs—smaller in size compared to recent million-scale datasets—it offers unique advantages in annotation quality, region-level detail, and diversity. Specifically, the dataset covers twelve major imaging categories, such as *Abdominal, Bone and Joint, Breast, Cardiac, Chest, Cranial, Dental, Dermatological, Endoscopy, Fundus, Gynecological, and Pathology Slide Imaging*, spanning thirty specialized disease categories. Each image is paired with both a global view and sev-

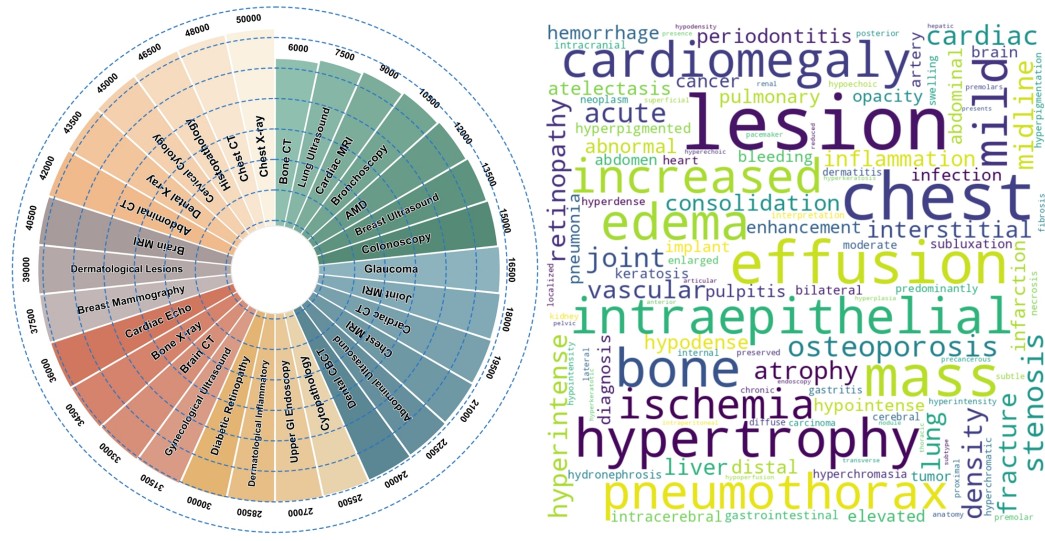

**a. Training image characterization**          **b. MedRegion-500k Word Cloud**

Figure 1: Overview of the MedRegion-500k dataset. (a) Distribution of medical images across major modalities and anatomical regions, highlighting the scale and diversity of our collection. (b) Word cloud of representative medical concepts and diagnostic terms, illustrating the semantic richness of the dataset.

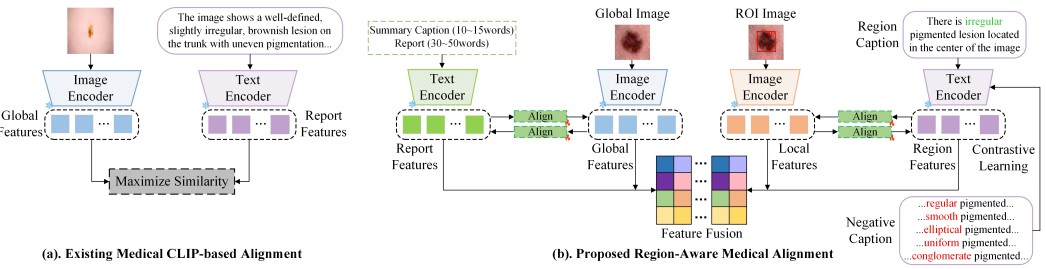

**(a). Existing Medical CLIP-based Alignment**          **(b). Proposed Region-Aware Medical Alignment**

Figure 2: Comparison between traditional global image-text alignment and our proposed region-aware contrastive learning framework. (a) Conventional CLIP-based methods align global image representations with textual descriptions, neglecting localized pathological features. (b) Our RegionMed-CLIP integrates global and region-specific features through a dedicated ROI processor, enabling more accurate and fine-grained alignment.

eral region-of-interest (ROI) crops, and is annotated with four types of textual descriptions: a summary caption, a detailed report caption, a region-specific caption, and multiple negative captions generated by perturbing the region description. High-quality ROI annotations are automatically generated using a pipeline that combines advanced detection and segmentation models, including Med-SAM Ma et al. (2024) and Grounding DINO Liu et al. (2023). Our annotation process employs a human-AI collaborative approach: we initially generate a small subset of annotations using Qwen-2.5VL-72B Bai et al. (2025), followed by careful human review and correction to ensure clinical accuracy. These human-validated annotations are then used to fine-tune the Qwen model via supervised fine-tuning (SFT) Ouyang et al. (2022); Wei et al. (2022), creating a specialized annotation model. This refined SFT model subsequently generates high-quality textual descriptions for the remaining dataset at scale, maintaining both consistency and clinical relevance.

Based on this dataset, we present RegionMed-CLIP, a region-aware multimodal contrastive learning framework designed for medical image understanding. Unlike traditional CLIP-based methods Radford et al. (2021); Wu et al. (2023), which focus primarily on global image features, RegionMed-

CLIP integrates both global and localized features by incorporating region-level information through a dedicated ROI processor. This processor enables fine-grained alignment between images and multi-level text, allowing the model to capture both the broader context and subtle, localized pathological cues Huang et al. (2021); Liu et al. (2024). As illustrated in Figure 2, conventional CLIP-based models (panel (a)) align global image representations with textual descriptions, but often overlook critical localized pathological features Zhang et al. (2023); Chen et al. (2024). In contrast, our proposed approach (panel (b)) incorporates both global and region-specific features, enhancing the accuracy and granularity of the image–text alignment Liu et al. (2024); Cui et al. (2024). Additionally, multiple negative captions are introduced for each region to further improve the model's discriminative capability, following recent advances in contrastive training for challenging negative mining in the medical domain Wang et al. (2024a); Gao et al. (2021). By progressively fusing global and local features and utilizing contrastive learning with challenging negatives, RegionMed-CLIP achieves higher model interpretability and recognition accuracy, particularly for detecting subtle and clinically relevant findings.

In summary, our contributions are as follows: (1) We propose RegionMed-CLIP, a region-aware multimodal contrastive learning framework for medical image understanding. It jointly encodes global and regional features with multi-level text and leverages an ROI processor for fine-grained vision–language alignment, achieving greater clinical relevance than global-only models. (2) We present MedRegion-500k, a carefully constructed medical image–text dataset. Though smaller in scale, it excels in annotation granularity, region diversity, and clinical coverage. Automated detection, segmentation, and language models ensure consistent, detailed region-level and textual annotations for each sample. (3) We establish a new benchmark by showing RegionMed-CLIP outperforms state-of-the-art models on image–text retrieval, zero-shot classification, and visual question answering. Our method achieves superior results with a smaller-scale dataset, highlighting the key role of high-quality, region-aware annotations.

## 2 RELATED WORK

**Vision-Language Pre-training in Medicine.** Recent advances in large-scale multimodal pre-training have fundamentally changed the landscape of computer vision and language processing Radford et al. (2021); Li et al. (2023); Liu et al. (2025). Pioneering models such as CLIP Radford et al. (2021), ALIGN Jia et al. (2021), and more recently MM-CLIP Liu et al. (2025), demonstrate that joint image-text representation learning enables strong zero-shot and transfer performance across domains. However, the direct transfer of these models to medicine is hindered by domain-specific terminology, strict privacy regulations, and the limited availability of expert-annotated data. To overcome these challenges, a range of medical adaptations have been proposed. For instance, BioViL Boecking et al. (2022) and MedCLIP Wang et al. (2022) utilize clinical reports to facilitate contrastive pre-training, while BiomedCLIP Huang et al. (2023) and PMC-CLIP Wu et al. (2023) scale up to tens of millions of medical image-text pairs, leveraging domain-specific encoders for improved alignment. Despite this progress, the majority of medical vision-language models remain focused on global alignment, often lacking explicit mechanisms for fine-grained pathological region modeling Liu et al. (2025).

**Region-Aware Multimodal Learning.** The integration of region-aware architectures has proven essential for both general and medical vision-language tasks He et al. (2017); Redmon et al. (2016); Yang et al. (2025). General-domain models such as Mask R-CNN He et al. (2017), YOLO Redmon et al. (2016), and recent adaptive region frameworks Yang et al. (2025) leverage spatial attention and region proposals to improve detection and localization. In multimodal reasoning, ViLBERT Lu et al. (2019) and UNITER Chen et al. (2020) introduce cross-modal attention mechanisms to better align textual and visual information. Within the medical domain, several approaches attempt to incorporate region-aware learning, though most are developed for general computer vision tasks and represent early attempts to explore region-aware capabilities in medical contexts. For instance, GLoRIA Huang et al. (2021) and BioViL-T Boecking et al. (2022) introduce hierarchical fusion modules for medical images, while REFERS Zhang et al. (2022) attempts region-level medical image understanding. However, models like MLLM Li et al. (2024), MAVL Phan et al. (2024), and Bannur et al. Bannur et al. (2023) are primarily designed for general-domain applications and have not been specifically optimized for medical region understanding. Similarly, approaches like Alpha-CLIP Sun et al. (2024) focus on general visual grounding rather than medical pathology localization.

Our work develops a region-aware framework specifically tailored for medical image understanding, addressing unique challenges of pathological region detection and clinical report alignment distinct from general-domain tasks. Nonetheless, many existing methods are constrained by the scarcity and limited diversity of region-annotated medical datasets, as well as reliance on external or handcrafted region proposals, which can hinder generalization in complex clinical settings Yang et al. (2025).

**Large-Scale Medical Multimodal Datasets.** High-quality annotated datasets are a prerequisite for robust medical AI Johnson et al. (2019); Irvin et al. (2019); Chen et al. (2025). Widely used resources such as MIMIC-CXR Johnson et al. (2019), CheXpert Irvin et al. (2019), and OpenI Demner-Fushman et al. (2016) offer paired images and reports, but do not include region-level labels. Recent initiatives such as PMC-15M Wu et al. (2023), MedPix MedPix (2020), and OpenMed-CLIP Chen et al. (2025) have significantly increased dataset scale, but still face limitations in regional granularity or modality diversity. The lack of comprehensive region-level annotations has constrained progress in fine-grained medical vision-language learning, underscoring the importance of new resources like MedRegion-500k, which combines broad modality coverage, curated ROI crops, and multi-level descriptions with rigorous quality control Chen et al. (2025).

**Progressive Training and Negative Mining.** Recent studies highlight the effectiveness of progressive and curriculum-based training strategies in multimodal learning Wang et al. (2024b); Zhang et al. (2025). Gradually increasing the complexity of supervision—from global alignment to region-level tasks—has been shown to improve model stability and generalization. Additionally, negative mining, especially using clinically similar but semantically distinct negatives, further enhances discriminative capability in both vision and medical tasks Wang et al. (2024b); Zhang et al. (2025). Nevertheless, the combined use of progressive multi-stage training and advanced negative mining at scale remains underexplored in most existing medical vision-language models, presenting an opportunity for future advancements Zhang et al. (2025).

## 3 METHODOLOGY

RegionMed-CLIP is a region-aware multimodal contrastive learning framework that combines global and local semantic cues for enhanced medical image understanding. The framework employs a dual-branch encoder processing both entire images and region-of-interest (ROI) crops, enabling simultaneous modeling of broad semantic context and localized disease-specific features.

**Framework Overview.** Unlike traditional CLIP-based methods that focus on global features alone, RegionMed-CLIP captures multi-scale associations through progressive fusion and staged supervision. The framework addresses the gap between coarse global visual-textual alignment and fine-grained localization of clinically relevant pathologies through an innovative ROI processor that explicitly integrates regional and global information. The overall architecture and training process are illustrated in Figure 3.

**Dual-Branch Image Encoding.** RegionMed-CLIP utilizes a ViT-B/16 image encoder $f_{\text{img}}(\cdot)$ to process both original images $x_{\text{global}}$ and ROI crops $x_{\text{roi}}$. ROI crops are automatically generated through Med-SAM and Grounding DINO detection pipeline. Visual embeddings are extracted as:

$$z_{\text{global}} = f_{\text{img}}(x_{\text{global}}), \quad z_{\text{roi}} = f_{\text{img}}(x_{\text{roi}}). \tag{1}$$

Here $z_{\text{global}}$ encodes the entire scene for high-level anatomical understanding, while $z_{\text{roi}}$ captures localized pathological details.

**Text Encoding and Multi-Level Annotations.** RegionMed-CLIP uses PubMedBERT as the text encoder $f_{\text{txt}}(\cdot)$ to process clinical annotations. Each medical image is paired with four annotation types: summary caption, detailed report caption, region-specific caption, and negative captions generated by perturbing clinical terms. For each textual description $t_j$, the embedding is computed as $h_j = f_{\text{txt}}(t_j)$. These hard negatives improve contrastive learning by enhancing discrimination between clinically similar pathologies.

**ROI Processor Architecture.** The ROI processor fuses global and regional features through multi-head cross-attention mechanisms. Given variable numbers of ROI crops per image, we employ padding and masking strategies to handle batching.

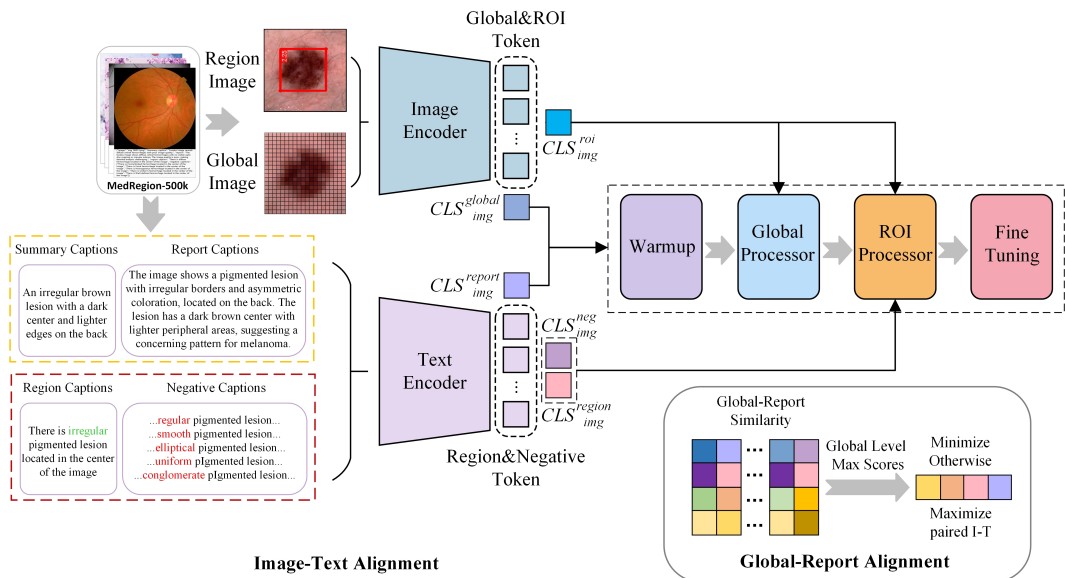

Figure 3: Architecture overview of RegionMed-CLIP. The framework progresses through four stages: (1) Warmup with global image-text alignment, (2) Global processing for holistic understanding, (3) ROI processing for region-aware feature fusion, and (4) Fine-tuning for joint optimization. The ROI processor integrates global semantic context with localized pathological details through sophisticated attention mechanisms.

The processor takes L2-normalized global features $\tilde{z}_{\text{global}}$ and ROI features $\tilde{z}_{\text{roi}}$ as input, computing attention weights between global context and each ROI region. Positional encodings are injected to preserve spatial relationships. The fused representation combines both global semantic understanding and fine-grained regional pathology detection capabilities. As shown in Figure 4, the ROI alignment matrix encourages matched region-text pairs to have high similarity scores, while the negative mining matrix reduces similarity to hard negatives. This dual-matrix approach supports robust region-level alignment and discriminative learning across diverse pathological presentations. The contrastive learning process effectively distinguishes between clinically relevant regional features and confounding background elements, enabling precise pathological localization that complements global image understanding.

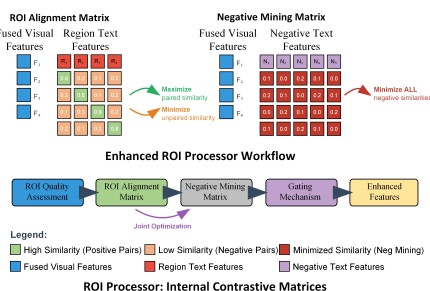

Figure 4: ROI processor contrastive learning matrices showing alignment optimization and negative mining processes.

**Progressive Multi-Stage Training.** Training follows a four-stage curriculum: warmup with global image-text alignment, global processing for semantic consistency, ROI processing with region-caption supervision, and joint fine-tuning of all components. This staged approach ensures stable convergence while progressively building from coarse to fine-grained understanding. L2 normalization is applied to all embeddings to operate on the unit hypersphere.

**Loss Functions and Optimization.** The training objective combines three weighted loss components:

$$\mathcal{L}_{\text{ROI}} = -\sum_i \log \frac{\exp(S_{i,i}/\tau)}{\sum_j \exp(S_{i,j}/\tau)}, \quad \mathcal{L}_{\text{neg}} = -\sum_{i,k} \log(1 - \sigma(N_{i,k})) \tag{2}$$

where $S_{i,j} = \tilde{z}_{\text{roi},i}^T \tilde{h}_{\text{region},j}$ represents ROI-caption alignment scores, $N_{i,k} = \tilde{z}_{\text{roi},i}^T \tilde{h}_{\text{neg},k}$ denotes negative mining scores, $\tau$ is the temperature parameter, and $\sigma(\cdot)$ is the sigmoid function. The ROI

alignment loss follows InfoNCE formulation while negative mining uses binary cross-entropy to penalize high similarity with hard negatives.

The global alignment loss is formulated as:

$$\mathcal{L}_{\text{global}} = -\sum_i \log \frac{\exp(\text{sim}(\tilde{z}_{\text{global},i}, \tilde{h}_{\text{report},i})/\tau)}{\sum_j \exp(\text{sim}(\tilde{z}_{\text{global},i}, \tilde{h}_{\text{report},j})/\tau)} \quad (3)$$

The unified training objective with learned weights is:

$$\mathcal{L}_{\text{total}} = \lambda_g \mathcal{L}_{\text{global}} + \lambda_r \mathcal{L}_{\text{ROI}} + \lambda_n \mathcal{L}_{\text{neg}} \quad (4)$$

where $\lambda_g$, $\lambda_r$, and $\lambda_n$ are loss weights optimized during training to balance global alignment, region-level precision, and negative mining objectives.

## 4 EXPERIMENTS AND ANALYSIS

This section evaluates RegionMed-CLIP across different medical image tasks, including zero-shot classification, VQA, and image-text retrieval. Comparisons with state-of-the-art models are presented, alongside an ablation study to assess the contributions of individual components.

### 4.1 EXPERIMENTAL SETUP

**Dataset Splits and Statistics.** Table 1 presents the detailed dataset splits used in our experiments across all evaluated medical datasets. Our MedRegion-500k dataset follows a standard 80/10/10 split with 400k training samples, 50k validation samples, and 50k test samples. For external evaluation datasets, we adhere to their official splits where available, or apply standard train/validation/test ratios when splits are not predefined. The diversity in dataset sizes ranges from smaller specialized collections like RFMiD2 (620 samples) to larger general datasets like PCam200 (47k samples), ensuring comprehensive evaluation across various medical imaging domains and data scales. All splits maintain class balance and avoid data leakage between training and evaluation phases, following standard practices in medical AI evaluation protocols.

| Dataset | Train/Valid/Test | Total |
|---|---|---|
| Ours | 400k/50k/50k | 500k |
| NLM-TB | 480/160/160 | 800 |
| SIIM-ACR | 2.7k/840/880 | 4.4k |
| LC25000 | 6.2k/2.1k/2.2k | 10.5k |
| Covid-CXTR2 | 12.8k/4.3k/4.5k | 21.6k |
| HyperKvasir | 5.6k/1.9k/2.0k | 9.5k |
| ODIR | 3.9k/1.3k/1.4k | 6.6k |
| PCam200 | 21.6k/7.2k/18.2k | 47k |
| RFMiD2 | 350/115/155 | 620 |
| MedFMC | 1.3k/435/450 | 2.2k |
| Breast Cancer | 2.4k/810/850 | 4.1k |
| VQA-RAD | 2.3k/766/451 | 3.5k |
| SLAKE | 3.7k/1.2k/1.1k | 6.0k |

Table 1: Dataset splits and statistics for all medical datasets used in our experiments.

**Dataset Example.** Figure 5 illustrates a representative sample from our MedRegion-500k dataset, showing both global and region-of-interest views of an abdominal CT scan. The left panel displays the original medical image, while the right panel highlights the identified pathological region with precise localization coordinates.

The corresponding multi-level textual annotations for this sample are structured as follows:

```
{"image": "img_00038.png", "summary_caption": "Liver shows well-defined
homogenous mass with slightly increased density.", "report": "In the
axial CT image, the liver demonstrates a well-defined, homogenous
mass with slightly increased density compared to surrounding liver
parenchyma, without evidence of fat stranding or perilesional edema.",
"region_caption": "There is homogenous mass located in the center of the
image", "negative_captions": ["There is Inhomogeneous mass located in the
center of the image", "There is Heterogeneous mass located in the center
of the image", "There is Non-uniform mass located in the center of the
image", "There is Irregular mass located in the center of the image",
"There is Varied mass located in the center of the image"]}
```

**Category: Abdominal | Type: Abnormal**

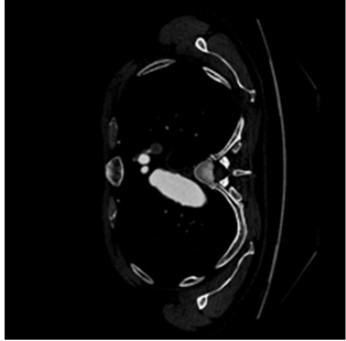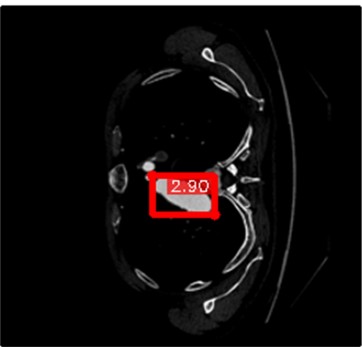

Figure 5: Example from MedRegion-500k dataset showing global image (left) and ROI annotation (right) for an abdominal CT scan with liver mass.

MedRegion-500k is constructed from diverse publicly available medical imaging resources and ethically approved datasets. The collection includes data from open platforms such as Kaggle (e.g., Pancreas dataset, Abdominal Ultrasound Images), established medical repositories like MIMIC-CXR that require institutional approval, and other clinical databases accessed through proper medical research agreements. All data usage strictly follows medical privacy regulations and ethical guidelines. Upon paper acceptance, we will release a curated subset of MedRegion-500k that complies with public sharing policies.

**Implementation Details.** All experiments are conducted using 4 NVIDIA A100 GPUs, demonstrating the method's practicality on modest computational resources. Images are resized to $224 \times 224$ pixels. The backbone comprises ViT-B/16 for image encoding and PubMedBERT Lee et al. (2020) for text encoding, both with 512-dimensional projections. The ROI processor incorporates 8 attention heads, and the contrastive loss temperature is set to 0.07. Feature vectors are L2-normalized Radford et al. (2021). Training proceeds in four stages: warmup (2 epochs, learning rate $1 \times 10^{-4}$, batch size 16), global alignment (8 epochs, $2 \times 10^{-4}$), region refinement (8 epochs, $2 \times 10^{-4}$), and joint fine-tuning (8 epochs, $3 \times 10^{-5}$, batch size 12). AdamW Loshchilov & Hutter (2018) is used with a weight decay of 0.02 and gradient clipping at 0.5 Pascanu et al. (2013). All input text is lowercased and stripped of punctuation. Evaluation covers diverse medical datasets: NLM-TB Jaeger et al. (2014), SIIM-ACR Zawacki et al. (2019), LC25000 Borkowski et al. (2019), Covid-CXTR2 Wang et al. (2020), HyperKvasir Borgli et al. (2020), ODIR Li et al. (2021), PCam200 Kawai et al. (2023), RFMiD2 Panchal et al. (2023), MedFMC Wang et al. (2023), Breast Cancer Hayder et al. (2024), MedRegion-500k, and VQA tasks on VQA-RAD and SLAKE Lau et al. (2022; 2018). Metrics include Recall@K for retrieval, AUC for classification, and accuracy for VQA, under both transductive and non-transductive evaluation protocols.

### 4.2 MAIN RESULTS

**Zero-Shot Classification.** RegionMed-CLIP is evaluated across ten medical datasets, as summarized in Table 2. The model consistently outperforms existing baselines across diverse medical imaging domains, with notable improvements including NLM-TB (90.89% vs. BiomedCLIP's 88.60%) and PCam200 (85.42% vs. BiomedCLIP's 83.16%). These results demonstrate the effectiveness of integrating global and localized features for medical image classification, particularly for detecting localized pathologies that contain vital diagnostic information.

**Medical Visual Question Answering (VQA).** RegionMed-CLIP is evaluated on VQA-RAD and SLAKE benchmarks, which assess clinical reasoning based on medical images. As shown in Table 3, RegionMed-CLIP achieves 83.6% overall accuracy, outperforming the strongest baseline SigLIP-400M (81.3%). The model shows particularly strong performance on SLAKE, reaching 90.1% overall accuracy and 92.1% on closed-ended questions. The ROI processor's explicit integration

| Dataset | CLIP | SigLIP-400M | PMC-CLIP | BiomedCLIP | Ours |
|---|---|---|---|---|---|
| NLM–TB | 65.64 | 82.92 | 74.22 | 88.60 | **90.89** |
| SIIM–ACR | 55.13 | 67.87 | 64.19 | 77.08 | **79.34** |
| LC25000 (COLON) | 66.53 | 88.31 | 98.78 | 98.89 | **98.91** |
| Covid–CXTR2 | 49.40 | 70.34 | 57.41 | 68.99 | **72.67** |
| HyperKvasir | 58.77 | 71.48 | 68.49 | 79.76 | **82.14** |
| ODIR | 52.29 | 67.44 | 67.20 | 69.61 | **71.72** |
| PCam200 | 59.87 | 71.87 | 79.44 | 83.16 | **85.42** |
| RFMiD2 | 42.47 | 45.71 | 41.40 | 41.11 | **47.05** |
| MedFMC (Chest) | 49.81 | 61.80 | 49.85 | 50.20 | **63.72** |
| Breast Cancer | 46.45 | 53.26 | 50.77 | 54.84 | **56.49** |

Table 2: AUC scores (%) for classification results across different medical datasets in the zero-shot setting. **Bold** indicates the best results and underline indicates the second best.

| Model | VQA-RAD | | | | SLAKE | | | | Overall |
|---|---|---|---|---|---|---|---|---|---|
| | Open | Closed | Overall | Avg | Open | Closed | Overall | Avg | |
| CLIP | 59.9 | 79.4 | 71.3 | 70.2 | 78.6 | 81.0 | 79.5 | 79.7 | 74.9 |
| PubMedCLIP | 60.1 | 80.0 | 72.1 | 70.7 | 78.4 | 82.5 | 80.1 | 80.3 | 75.5 |
| SigLIP-400M | 68.5 | 79.5 | 75.2 | 74.4 | 85.8 | 90.3 | 88.5 | 88.2 | 81.3 |
| BiomedCLIP | 67.0 | 76.5 | 72.7 | 72.1 | 84.3 | 88.9 | 86.1 | 86.4 | 79.3 |
| **Ours** | **70.4** | **82.3** | **77.4** | **76.8** | **87.5** | **92.1** | **90.1** | **90.4** | **83.6** |

Table 3: Comprehensive accuracy (%) comparison on VQA-RAD and SLAKE datasets across different question types. **Bold** indicates the best results and underline indicates the second best.

of regional features enables more precise localization of clinically important cues, confirming that region-aware alignment improves medical VQA performance.

**Image-Text Retrieval.**

Image-text retrieval experiments demonstrate RegionMed-CLIP's effectiveness in associating medical images with their corresponding textual descriptions. As presented in Table 4, RegionMed-CLIP achieves 49.7% Recall@1 for text-to-image and 50.3% for image-to-text retrieval, outperforming the best baseline SigLIP-400M (45.4% and 47.8% respectively). The consistent improvements across all recall metrics highlight the value of integrating both global and region-

| Model | T2I | | | I2T | | |
|---|---|---|---|---|---|---|
| | R@1 | R@5 | R@10 | R@1 | R@5 | R@10 |
| CLIP | 32.9 | 58.6 | 72.4 | 35.2 | 61.8 | 75.7 |
| PMC-CLIP | 38.8 | 64.9 | 78.5 | 41.1 | 67.2 | 80.8 |
| BiomedCLIP | 42.7 | 68.8 | 81.4 | 44.9 | 71.1 | 83.6 |
| SigLIP | 45.4 | 71.5 | 83.1 | 47.8 | 73.9 | 85.4 |
| **Ours** | **49.7** | **74.2** | **85.1** | **50.3** | **75.8** | **86.3** |

Table 4: Performance comparison on medical image-text retrieval (%). **Bold** indicates best results, underline indicates second best.

specific features for robust medical image-text alignment. Figure 6 demonstrates the attention localization capabilities of RegionMed-CLIP across diverse medical imaging modalities. The visualization reveals that our model successfully identifies and focuses on pathologically relevant regions: in fundus imaging, attention concentrates on retinal lesions and vascular abnormalities; in dermatological images, the model precisely localizes skin lesions; dental imaging shows focused attention on tooth structures and potential anomalies; and abdominal CT scans highlight anatomically signif-

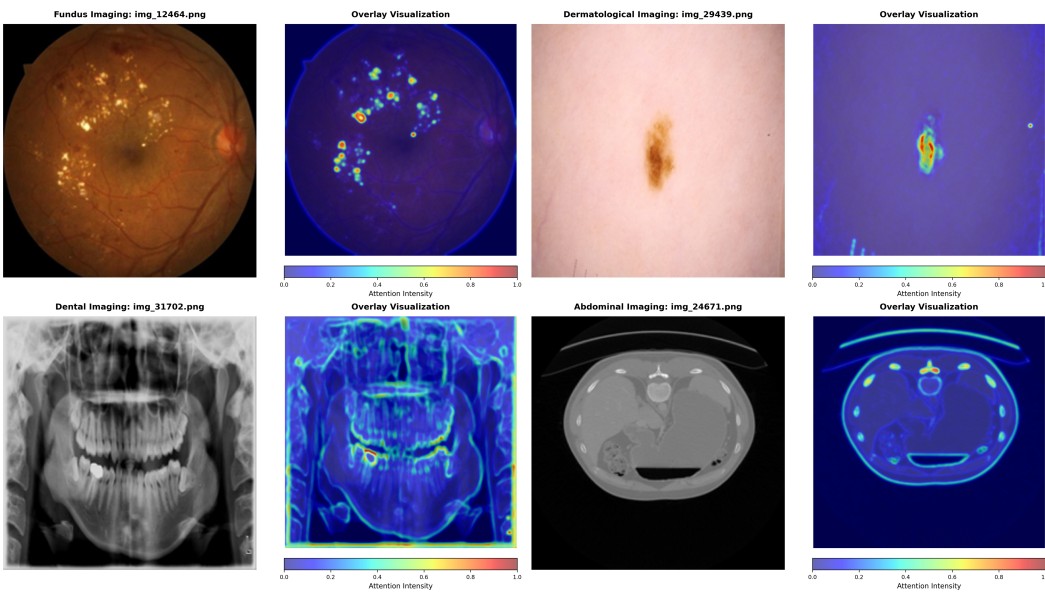

Figure 6: Attention visualization across different medical imaging modalities. Left columns show original images, right columns display attention heatmaps. RegionMed-CLIP successfully localizes pathologically relevant regions across diverse modalities.

icant regions. This attention-based localization validates that RegionMed-CLIP effectively captures fine-grained pathological features that are critical for clinical interpretation.

**Ablation Studies.**

To assess individual component contributions, we conduct systematic ablation studies by progressively adding modules. Figure 7 shows results for: (1) baseline image-text contrastive model, (2) warmup with global alignment, (3) global processing, (4) ROI processing, and (5) joint fine-tuning. The study reveals key insights: global processing significantly enhances performance with Recall@1 increasing from 15.2% to 32.4%, demonstrating that global se-

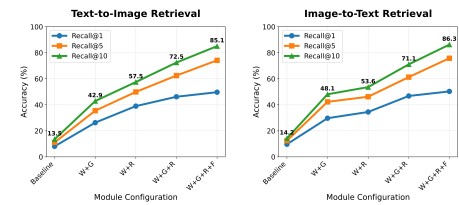

Figure 7: Ablation study showing progressive improvements with each component addition.

mantic features are essential for medical image understanding. The most substantial improvement comes from ROI processing, increasing Recall@1 to 41.8%, emphasizing the importance of fine-grained, region-specific learning in medical tasks where pathology is localized. Joint fine-tuning of all components leads to optimal performance at 43.7%, confirming that optimizing both global and region-level features together is crucial for achieving the best results. Each component contributes meaningfully to the overall system performance.

## 5 CONCLUSION

This paper presents RegionMed-CLIP, a novel region-aware multimodal contrastive learning framework for medical image understanding. By integrating global and region-specific features, RegionMed-CLIP improves detection of localized pathologies often overlooked by traditional models. Extensive experiments on image-text retrieval, zero-shot classification, and visual question answering show promising results compared to existing models. The MedRegion-500k dataset with detailed regional annotations further contributes to these improvements. These results validate the importance of region-aware multimodal alignment and establish RegionMed-CLIP as a promising foundation for medical image processing and clinical decision support.

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

## A   APPENDIX

## B   ETHICS STATEMENT

This work adheres to the ICLR Code of Ethics. In this study, no human subjects or animal experimentation was involved. All datasets used, including MedRegion-500k, were sourced in compliance with relevant usage guidelines, ensuring no violation of privacy. We have taken care to avoid any biases or discriminatory outcomes in our research process. No personally identifiable information was used, and no experiments were conducted that could raise privacy or security concerns. We are committed to maintaining transparency and integrity throughout the research process.

## C   REPRODUCIBILITY STATEMENT

We have made every effort to ensure that the results presented in this paper are reproducible. The code implementation will be provided in the supplementary materials to facilitate replication and verification. The experimental setup, including training steps, model configurations, and hardware details, is described in detail in the paper.

Our MedRegion-500k dataset is constructed from diverse publicly available medical imaging resources and ethically approved datasets. The collection includes data from open platforms such as Kaggle (e.g., Pancreas dataset, Abdominal Ultrasound Images), established medical repositories like MIMIC-CXR that require institutional approval, and other clinical databases accessed through proper medical research agreements. All data usage strictly follows medical privacy regulations and ethical guidelines. Upon paper acceptance, we will release a curated subset of MedRegion-500k that complies with public sharing policies.

Additionally, other public datasets used in our experiments are publicly available, ensuring consistent and reproducible evaluation results.

We believe these measures will enable other researchers to reproduce our work and further advance the field.

## D   LLM USAGE

Large Language Models (LLMs) were used to aid in the writing and polishing of the manuscript. Specifically, we used an LLM to assist in refining the language, improving readability, and ensuring clarity in various sections of the paper. The model helped with tasks such as sentence rephrasing, grammar checking, and enhancing the overall flow of the text.

It is important to note that the LLM was not involved in the ideation, research methodology, or experimental design. All research concepts, ideas, and analyses were developed and conducted by the authors. The contributions of the LLM were solely focused on improving the linguistic quality of the paper, with no involvement in the scientific content or data analysis.

The authors take full responsibility for the content of the manuscript, including any text generated or polished by the LLM. We have ensured that the LLM-generated text adheres to ethical guidelines and does not contribute to plagiarism or scientific misconduct.

## E  NOTATIONS

| Symbol | Explanation |
|---|---|
| $x_{global}$ | Global input medical image |
| $x_{roi}$ | Region-of-interest crop from medical image |
| $z_{global}$ | Global image embedding |
| $z_{roi}$ | ROI image embedding |
| $f_{img}(\cdot)$ | Image encoder function |
| $f_{txt}(\cdot)$ | Text encoder function |
| $t_j$ | Textual description (caption or report) |
| $h_j$ | Text embedding for description $t_j$ |
| $F_i$ | Normalized ROI feature embedding |
| $R_j$ | Normalized region caption embedding |
| $R_{neg}(k)$ | Negative caption embedding |
| $S_{i,j}$ | Similarity score between ROI $i$ and caption $j$ |
| $N_{i,k}$ | Negative mining similarity score |
| $\mathcal{L}_{global}$ | Global alignment loss |
| $\mathcal{L}_{roi}$ | ROI alignment loss |
| $\mathcal{L}_{neg}$ | Negative mining loss |
| $\mathcal{L}_{total}$ | Total combined loss |
| $\tau$ | Temperature for contrastive learning |
| $\sigma(\cdot)$ | Sigmoid activation function |
| $\mathcal{D}_{train}$ | Training dataset |
| $\mathcal{D}_{val}$ | Validation dataset |
| $\mathcal{D}_{test}$ | Test dataset |
| $\lambda_g, \lambda_r, \lambda_n$ | Loss weighting parameters |
| $\tilde{z}_{global}$ | L2-normalized global image embedding |
| $\tilde{z}_{roi}$ | L2-normalized ROI image embedding |
| $\tilde{h}_{region}$ | L2-normalized region text embedding |
| $\tilde{h}_{report}$ | L2-normalized report text embedding |
| $\tilde{h}_{neg}$ | L2-normalized negative text embedding |

Table 5: Notations used in this paper.

## F  B. PROMPT FOR QWEN2.5VL-72B

For the automated annotation pipeline utilizing Qwen2.5VL-72B, carefully designed prompts are employed to synthesize clinical expertise with rigorous annotation criteria. These prompts are structured to ensure that medical image descriptions remain consistent, clinically accurate, and diagnostically relevant across a wide range of imaging modalities.

**System Prompt Design:** The system prompt positions the model as an experienced radiologist with expertise in interpreting chest X-rays, CT scans, MRI, endoscopy, pathology slides, and other diagnostic modalities. This structured strategy is adopted to guarantee comprehensive and clinically meaningful annotations for twelve major imaging modalities, including radiological, pathological, and endoscopic images.

**Summary Generation:** The summary caption is required to be a single, diagnostic sentence consisting of 10–15 words, without commas or semicolons, and ending with a period. Each summary is expected to highlight the principal finding and anatomical location in clear, simple language, focusing on observations directly relevant to clinical diagnosis.

**Region Extraction:** For each medical imaging report, exactly three components are extracted: (1) a shape or appearance adjective (e.g., irregular, round, oval, raised), (2) a medical lesion term (e.g.,

---

**Medical Image Analysis Prompt**

**Role:** Experienced radiologist analyzing medical images
**Task:** Generate structured JSON annotation with clinical accuracy
**JSON Format:**

```
{
    "image": "filename.png",
    "summary_caption":
          "Brief diagnostic summary (10-15 words)",
    "report":
          "Detailed clinical findings report",
    "region_caption":
          "Complete sentence describing ROI",
    "negative_captions":
          ["5 contrasting alternatives"]
}
```

**Specific Guidelines:**

- **summary_caption**: Single sentence, 10-15 words, highlights principal finding and anatomical location

- **report**: Comprehensive description including imaging modality, anatomical structure, abnormality characteristics, and clinical context

- **region_caption**: Complete sentence starting with "There is [adjective] [pathology] located in [position]"

- **negative_captions**: Generate 5 alternatives by systematically replacing key descriptive adjectives with contrasting terms (e.g., homogenous→heterogeneous, well-defined→irregular, uniform→varied)

- Use standard medical terminology consistently

- Positions: center/left/right, upper/lower/middle, or anatomical combinations

- If no abnormalities visible, use "No Finding" for all fields

---

Figure 8: Structured prompt for medical image annotation

lesion, mass, nodule, opacity), and (3) an anatomical location, mapped to standardized categories such as left, right, center, central, upper, lower, and their combinations (e.g., left upper, right lower).

**Negative Sample Generation:** Five alternative shape or appearance adjectives are systematically generated to replace the original descriptor, prioritizing contrasting characteristics. This approach introduces meaningful negative samples, which facilitate robust contrastive learning by systematically varying texture, color, shape, or location.

**Clinical Standards and Output:** All annotations are generated with an emphasis on describing only clearly visible abnormalities, avoiding speculation, and employing standardized medical terminology and anatomical references. Image quality is also assessed where relevant, and objectivity is maintained throughout the process. The structured output combines clinically relevant information in a machine-readable format, providing the necessary depth and precision for the development and evaluation of robust medical vision–language models.

## G ONE-SHOT AND FEW-SHOT CLASSIFICATION

The one-shot and few-shot learning capabilities of RegionMed-CLIP are assessed to demonstrate its rapid adaptation to novel medical imaging tasks.

| Dataset | CLIP | SigLIP | PMC | Biomed | RegionMed-CLIP |
|---|---|---|---|---|---|
| NLM-TB | 68.12 | 85.34 | 76.85 | 90.22 | **94.67** |
| SIIM-ACR | 57.89 | 70.45 | 67.31 | 79.84 | **85.42** |
| LC25000 | 68.95 | 90.12 | 99.01 | 99.15 | **99.51** |
| Covid-CXTR2 | 52.18 | 73.67 | 60.12 | 71.45 | **84.23** |
| HyperKvasir | 61.23 | 74.12 | 71.08 | 82.14 | **84.18** |
| ODIR | 54.87 | 69.88 | 69.75 | 72.13 | **75.84** |
| PCam200 | 62.45 | 74.32 | 82.11 | 85.73 | **94.89** |
| RFMiD2 | 44.92 | **48.15** | 43.87 | 43.54 | 47.23 |
| MedFMC | 52.34 | 64.21 | 52.41 | 52.78 | **69.12** |
| Breast Cancer | 48.92 | 55.71 | 53.28 | 57.31 | **57.89** |
| **Average** | 57.19 | 70.58 | 67.58 | 73.41 | **79.30** |

Table 6: AUC scores (%) for one-shot classification. **Bold** indicates best, underline indicates second best.

## G.1 FEW-SHOT LEARNING RESULTS

Few-shot learning performance, evaluated using five labeled examples per class, is summarized in Table 7. RegionMed-CLIP demonstrates marked improvement in this setting, achieving an average AUC of 82.15%. These results suggest that the model effectively leverages limited supervision to enhance generalization across diverse medical imaging tasks.

| Dataset | CLIP | SigLIP | PMC | Biomed | RegionMed-CLIP |
|---|---|---|---|---|---|
| NLM-TB | 71.28 | 88.15 | 79.67 | 92.34 | **96.21** |
| SIIM-ACR | 60.45 | 73.21 | 70.89 | 82.67 | **87.93** |
| LC25000 | 71.87 | 92.45 | 99.23 | 99.34 | **99.67** |
| Covid-CXTR2 | 55.12 | 76.89 | 63.45 | 74.23 | **86.84** |
| HyperKvasir | 64.51 | 77.23 | 74.12 | 84.89 | **86.95** |
| ODIR | 57.84 | 72.67 | 72.31 | 74.78 | **78.45** |
| PCam200 | 65.78 | 77.89 | 85.43 | 88.92 | **96.87** |
| RFMiD2 | 47.83 | **51.24** | 46.78 | 46.23 | 50.15 |
| MedFMC | 55.67 | 67.45 | 55.23 | 55.89 | **72.34** |
| Breast Cancer | 52.18 | 58.94 | 56.45 | 60.12 | **61.08** |
| **Average** | 60.25 | 73.61 | 70.36 | 75.94 | **82.15** |

Table 7: AUC scores (%) for few-shot classification (5 examples per class). **Bold** indicates best, underline indicates second best.

A clear trend of progressive improvement is observed as additional labeled examples are incorporated. In the zero-shot setting, RegionMed-CLIP attains an average AUC of 77.09%. With one labeled example per class, the average AUC rises to 79.30%, and further increases to 82.15% when five examples per class are provided. These results underscore the model's efficiency in leverag-

ing limited supervision to boost performance, highlighting robust adaptability and superior sample efficiency across diverse medical classification tasks.

## H MATHEMATICAL DERIVATIONS

This section provides detailed mathematical derivations for the key components of RegionMed-CLIP, including the contrastive loss functions, ROI processor architecture, and optimization objectives.

### H.1 CONTRASTIVE LEARNING FRAMEWORK

**InfoNCE Loss Derivation.** The foundational contrastive loss in RegionMed-CLIP follows the InfoNCE framework. For a batch of $N$ samples, where each sample $i$ has a positive pair and $N - 1$ negative pairs, the InfoNCE loss maximizes the similarity between positive pairs while minimizing similarity to negatives.

Starting from the mutual information objective, we aim to maximize:

$$I(X;Y) = \mathbb{E}_{p(x,y)} \left[ \log \frac{p(x,y)}{p(x)p(y)} \right] \tag{5}$$

Using the contrastive estimation approach, this becomes:

$$\mathcal{L}_{\text{InfoNCE}} = -\mathbb{E} \left[ \log \frac{e^{f(x,y^+)/\tau}}{\sum_{j=1}^{N} e^{f(x,y_j)/\tau}} \right] \tag{6}$$

where $f(x, y) = \cos(\tilde{x}, \tilde{y})$ is the cosine similarity between L2-normalized embeddings, and $\tau$ is the temperature parameter.

**Global Alignment Loss.** For global image-text alignment, we apply InfoNCE to global image embeddings $\tilde{z}_{global,i}$ and report text embeddings $\tilde{h}_{report,j}$:

$$\mathcal{L}_{global} = -\frac{1}{N} \sum_{i=1}^{N} \log \frac{\exp(\tilde{z}_{global,i}^T \tilde{h}_{report,i}/\tau)}{\sum_{j=1}^{N} \exp(\tilde{z}_{global,i}^T \tilde{h}_{report,j}/\tau)} \tag{7}$$

The symmetric formulation includes both image-to-text and text-to-image directions:

$$\mathcal{L}_{global}^{sym} = \frac{1}{2}(\mathcal{L}_{global}^{i2t} + \mathcal{L}_{global}^{t2i}) \tag{8}$$

### H.2 REGION-AWARE CONTRASTIVE LEARNING

**ROI Alignment Loss.** For region-level alignment, we extend InfoNCE to handle multiple ROI crops per image. Let $\mathcal{R}_i$ denote the set of ROI crops for image $i$, and $r_{i,k}$ represent the $k$-th ROI crop. The ROI alignment loss becomes:

$$\mathcal{L}_{roi} = -\frac{1}{N} \sum_{i=1}^{N} \frac{1}{|\mathcal{R}_i|} \sum_{k \in \mathcal{R}_i} \log \frac{\exp(S_{i,k,i}/\tau)}{\sum_{j=1}^{N} \exp(S_{i,k,j}/\tau)} \tag{9}$$

where $S_{i,k,j} = \tilde{z}_{roi,i,k}^T \tilde{h}_{region,j}$ represents the similarity between the $k$-th ROI of image $i$ and the region caption of image $j$.

**Negative Mining Loss.** To enhance discriminative learning, we introduce hard negative mining using semantically similar but incorrect region descriptions. The negative mining loss employs binary cross-entropy:

$$\mathcal{L}_{neg} = -\frac{1}{N} \sum_{i=1}^{N} \frac{1}{|\mathcal{R}_i|} \sum_{k \in \mathcal{R}_i} \frac{1}{K} \sum_{m=1}^{K} \log(1 - \sigma(N_{i,k,m})) \tag{10}$$

where $N_{i,k,m} = \tilde{z}_{roi,i,k}^T \tilde{h}_{neg,i,m}$ is the similarity between ROI $k$ of image $i$ and its $m$-th negative caption, $K$ is the number of negative samples per ROI, and $\sigma(\cdot)$ is the sigmoid function.

The gradient of the negative mining loss with respect to ROI embeddings is:

$$\frac{\partial \mathcal{L}_{neg}}{\partial \tilde{z}_{roi,i,k}} = \frac{1}{K} \sum_{m=1}^{K} \sigma(N_{i,k,m}) \tilde{h}_{neg,i,m} \tag{11}$$

### H.3 ROI PROCESSOR MATHEMATICAL FRAMEWORK

**Cross-Attention Mechanism.** The ROI processor employs multi-head cross-attention to fuse global and regional features. For each ROI crop, the attention mechanism computes:

$$\text{Attention}(Q, K, V) = \text{softmax}\left(\frac{QK^T}{\sqrt{d_k}}\right) V \tag{12}$$

$$\text{MultiHead}(Q, K, V) = \text{Concat}(\text{head}_1, \ldots, \text{head}_h) W^O \tag{13}$$

where $\text{head}_i = \text{Attention}(QW_i^Q, KW_i^K, VW_i^V)$.

For ROI processing, we set:

$$Q = \tilde{z}_{roi} W^Q \quad \text{(ROI features as queries)} \tag{14}$$

$$K = \tilde{z}_{global} W^K \quad \text{(Global features as keys)} \tag{15}$$

$$V = \tilde{z}_{global} W^V \quad \text{(Global features as values)} \tag{16}$$

**Feature Fusion.** The final ROI representation combines attended global context with original ROI features:

$$\tilde{z}_{roi}^{fused} = \text{LayerNorm}(\tilde{z}_{roi} + \text{MultiHead}(\tilde{z}_{roi}, \tilde{z}_{global}, \tilde{z}_{global})) \tag{17}$$

### H.4 TOTAL LOSS AND OPTIMIZATION

**Weighted Loss Combination.** The total training objective combines all loss components with learnable weights:

$$\mathcal{L}_{total} = \lambda_g \mathcal{L}_{global} + \lambda_r \mathcal{L}_{roi} + \lambda_n \mathcal{L}_{neg} \tag{18}$$

To ensure balanced optimization, the loss weights are dynamically adjusted using uncertainty-based weighting:

$$\lambda_i = \frac{\exp(-s_i^2)}{2s_i^2}, \quad i \in \{g, r, n\} \tag{19}$$

where $s_i$ are learnable parameters representing the uncertainty of each loss component.

**Gradient Flow Analysis.** The gradient of the total loss with respect to image encoder parameters $\theta_{img}$ is:

$$\frac{\partial \mathcal{L}_{total}}{\partial \theta_{img}} = \lambda_g \frac{\partial \mathcal{L}_{global}}{\partial \theta_{img}} + \lambda_r \frac{\partial \mathcal{L}_{roi}}{\partial \theta_{img}} + \lambda_n \frac{\partial \mathcal{L}_{neg}}{\partial \theta_{img}} \tag{20}$$

$$= \lambda_g \frac{\partial \mathcal{L}_{global}}{\partial \tilde{z}_{global}} \frac{\partial \tilde{z}_{global}}{\partial \theta_{img}} + (\lambda_r + \lambda_n) \frac{\partial (\mathcal{L}_{roi} + \mathcal{L}_{neg})}{\partial \tilde{z}_{roi}} \frac{\partial \tilde{z}_{roi}}{\partial \theta_{img}} \tag{21}$$

This formulation ensures that gradients from both global and regional objectives contribute to encoder parameter updates, enabling joint optimization of multi-scale features.

**Temperature Parameter Analysis.** The temperature parameter $\tau$ controls the sharpness of the probability distribution in contrastive learning. Its optimal value can be derived by analyzing the bias-variance trade-off:

$$\tau^* = \arg \min_{\tau} \mathbb{E}[(\hat{\theta}_\tau - \theta^*)^2] \tag{22}$$

where $\hat{\theta}_\tau$ is the parameter estimate using temperature $\tau$ and $\theta^*$ is the true parameter. Empirically, we found $\tau = 0.07$ provides optimal performance across medical datasets.

