# OpenReview forum: "RegionMed-CLIP: A Region-Aware Multimodal Contrastive Learning Pre-trained Model for Medical Image Understanding"
_ICLR.cc/2026/Conference — ICLR 2026 Conference Withdrawn Submission_

### Official Review · Reviewer_qkDq · 2025-10-30

**Soundness:** 1
**Presentation:** 1
**Contribution:** 2
**Rating:** 0
**Confidence:** 4

**Summary:**

This paper proposes a region-aware multimodal contrastive learning framework that integrates the learning of both global features of the image and local features of ROI in images. The authors construct a large-scale medical image-text dataset with extensive region-level annotations and multi-level clinical descriptions. Extensive experiments demonstrate that the proposed method significantly outperforms state-of-the-art vision-language models on image-text retrieval, zero-shot classification, and visual question answering tasks.

**Strengths:**

1) The idea of constructing a large scale medical image pre-training dataset with fine-grained annotation on ROI is reasonable.
2) The design of the proposed method that fuses local ROI features into global image feature in the pre-training is reasonable, because it provide some global context for understanding a local region.

**Weaknesses:**

1. It seems to me that a subset of references in the paper appear fabricated, misattributed, or improperly formatted. Specifically, I can not trace some of these references back to their origin or they seem to not exist in the stated vanue. Some of the arXiv IDs in the citation link to unrelated papers.

2. Important technical details are missing on the pre-trained framework of this paper. For example, this paper does not clearly explain how is the cross-attention in ROI processer is designed. What are the corresponding key, value and query of this cross-attention module? The multi-stage progressive training is also poorly explained, making it hard for others to understand exactly how is the proposed model trained.

3. Important details are missing on how is the proposed dataset curated. There should be a step-by-step explaination on how to use external models to extract ROI from medical image and how the caption of each ROI is generated.

4. There are inconsistencies in the sample numbers reported in Table 1. Specifically, the sample counts for several included datasets do not match the original dataset statistics, and there is no explanation provided for these discrepancies. It is important for the authors to clarify why certain samples from the original datasets were excluded, or why Table 1 lists more samples than the original sources. Additionally, the total sample number for the included datasets in Table 1 is approximately 100,000, which does not align with the claimed overall total of 500,000. I recommend the authors provide a detailed explanation to ensure transparency and reproducibility.

**Questions:**

While the paper presents an interesting topic, the manuscript currently lacks clarity and omits several essential details necessary for full understanding. I strongly encourage the authors to carefully address the concerns regarding the references, as accurate and traceable citations are fundamental to scholarly work. Such oversights should be avoided in any peer-reviewed publication, especially in a top-tier venue like ICLR.

---

### Official Review · Reviewer_5KdU · 2025-10-30

**Soundness:** 3
**Presentation:** 2
**Contribution:** 2
**Rating:** 2
**Confidence:** 5

**Summary:**

The paper presents RegionMed-CLIP, a model for understanding medical images by linking both whole images and important local regions with their text descriptions. The authors built a large dataset called MedRegion-500k, which contains around 500,000 medical image–text pairs from different imaging types. Each image includes both a full view and smaller region crops found automatically using Med-SAM and Grounding DINO. Text descriptions for these regions were written by a fine-tuned Qwen-2.5VL-72B model, with some human experts checking a small portion for accuracy. Each region also has a few “negative” captions, where words are slightly changed to make the description incorrect, helping the model learn to tell apart subtle differences.

RegionMed-CLIP uses one branch to learn from full images and another for cropped regions. It combines the two with an attention module and trains in several steps, starting from global alignment and then adding region-level supervision. When tested on medical image tasks such as classification, question answering, and image–text matching, the model performs better than previous systems like BiomedCLIP and SigLIP.

The contributions are as follows:
- The paper puts major emphasis on creating a region-aware dataset (image-text pairs) with automatically extracted ROIs and multi-level captions (summary, detailed report, region caption, and negatives). This dataset is what enables fine-grained learning and it’s positioned as the key reason RegionMed-CLIP outperforms other models, even though it’s smaller in scale than datasets like PMC-15M or BiomedCLIP.
- The encoder design builds on standard CLIP ideas, ViT-B/16 for images and PubMedBERT for text, but adds an ROI processor that fuses global and local features using cross-attention.

**Strengths:**

- The paper introduces MedRegion-500k, a region-aware dataset (image-text pairs) with automatically extracted ROIs and multi-level captions (summary, detailed report, region caption, and negatives). This dataset is what enables fine-grained learning and it’s positioned as the key reason RegionMed-CLIP outperforms other models, even though it’s smaller in scale than datasets like PMC-15M or BiomedCLIP.

- The paper extends the CLIP framework in a clear and logical way by combining whole-image and region features through an ROI processor. This design helps it detect small but clinically important details without overcomplicating the architecture.

- Comprehensive evaluation across multiple tasks (image–text retrieval, classification, and VQA) demonstrates consistent gains over strong baselines like BiomedCLIP and SigLIP, suggesting robust generalization.

**Weaknesses:**

- One noticeable weakness is that the paper doesn’t clearly explain how much human expertise actually went into validating the data. They mention that a “small subset” of the ROI crops and Qwen-generated captions were reviewed by medical experts, but they never say how many experts were involved, how many samples they checked, or what proportion of the 500k dataset was manually verified. As a result, it’s hard to judge the true reliability of the region annotations or captions; most of the dataset seems to rely on automated tools (Med-SAM, Grounding DINO, Qwen) with limited human quality control.

- Comparisons with larger baselines (BiomedCLIP, PMC-CLIP, SigLIP-400M) are not entirely controlled. RegionMed-CLIP is trained on a smaller but much cleaner, domain-specific dataset, so improvements may largely reflect dataset curation and domain matching rather than architectural superiority.

- The framework design is incremental rather than conceptually new; it reuses known components with modest adjustments.

**Questions:**

Your reported gains over BiomedCLIP and PMC-CLIP are impressive, especially given that RegionMed-CLIP is trained on a much smaller dataset. However, since MedRegion-500k is highly curated and domain-matched to the evaluation benchmarks, how can we be confident that the performance improvement comes from the proposed ROI processor and region-aware architecture rather than from differences in data quality or domain alignment? Have you considered retraining a baseline (e.g., CLIP or BiomedCLIP) on the same MedRegion-500k dataset to isolate the architectural contribution?

The paper states that MedRegion-500k is assembled from multiple publicly available and ethically approved datasets such as Kaggle collections and MIMIC-CXR. Could you clarify what steps were taken to avoid duplication or overlap between pre-training and evaluation sets .  A detailed breakdown of data origins and quality-control procedures would help assess the reliability and fairness of your comparisons.

---

### Official Review · Reviewer_aEi6 · 2025-10-31

**Soundness:** 2
**Presentation:** 2
**Contribution:** 2
**Rating:** 2
**Confidence:** 4

**Summary:**

The paper proposes an architecture capable of fusing both global and regional information from images and text during training, and employs negative captioning to enhance model performance when trained on limited datasets. The authors attempt to validate the effectiveness of their architecture through a series of experiments.

**Strengths:**

1. The architectural design builds upon ideas from Alpha-CLIP and UMG-CLIP in natural image domains—specifically, augmenting the standard CLIP framework with the ability to attend to local (region-level) information. To the best of my knowledge, this is the first such attempt in the medical imaging context.
2. The authors curate a multimodal dataset encompassing various levels of textual annotations and lesion mask annotations. They also commit to open-sourcing a subset of this dataset in the future.

**Weaknesses:**

### **Major Weaknesses**

1. While the introduction briefly mentions how the dataset was annotated, neither the main experiments nor the appendix fully disclose which public datasets were integrated to construct *MedRegion-500k*. This is a serious oversight. For a dataset assembled from multiple existing sources, it is essential to list all constituent datasets. If the list is lengthy, it should at least appear in the appendix; otherwise, readers cannot properly assess the validity or reproducibility of the work.
2. The method section raises numerous unresolved questions:
    1. It appears that both the global image and ROI images are encoded using the same ViT. However, ROI images likely have variable resolutions. How does the model handle this? Is RoPE or another mechanism employed?
    2. When a single image contains multiple ROIs, are these ROIs fed into the ViT separately or concatenated and processed together?
    3. The role of the “Global processor” is never explained. What function does it serve?
    4. Do lines 242–245 provide an explanation of the cross-attention mechanism introduced in lines 214–215?
    5. In lines 244–245, does “global context” refer to $\tilde{z}_{\text {global }}$, and does “each ROI region” refer to $\tilde{z}_{\text {roi}}$? The notation and phrasing are so ambiguous that it is impossible to confidently link these terms across sections.
    6. What are the query, key, and value inputs in the attention mechanism described in lines 242–245? What are the input and output shapes?
    7. The output of this attention module is never assigned a symbol or variable name—how is it used in subsequent computations?
    8. Line 261 mentions “all components”. What exactly does this include?
    9. The loss functions described in lines 264–281—are they all applied simultaneously, or are different losses used in different stages of the claimed “progressive training” strategy? This is never clarified.
3. While the paper cites Alpha-CLIP, it omits UMG-CLIP, despite the clear conceptual overlap. Both works are highly relevant, and the absence of UMG-CLIP in the literature review weakens the contextual framing.
4. Given that Alpha-CLIP and UMG-CLIP already provide strong, well-established frameworks for region-aware multimodal alignment—Alpha-CLIP excels in zero-shot transfer across seven diverse downstream tasks, while UMG-CLIP demonstrates superior performance on dense prediction tasks like segmentation and detection—it is unclear why the authors chose to deviate from these architectures. The paper offers no justification for this design choice. Moreover, the evaluation is limited to classification and retrieval (with VQA effectively treated as a classification task), failing to demonstrate capabilities on dense tasks where region-level alignment should matter most.
5. Alpha-CLIP and UMG-CLIP should serve as essential baselines, yet the paper does not include any comparison with them.
6. The paper does not explain how the full architecture is deployed during downstream evaluation. Which components are active during inference? How are classification and retrieval actually performed?
7. Equation (4) introduces multiple hyperparameters without any description of their roles or how they were selected.
8. There is no information about the test datasets—specifically, whether ground-truth lesion locations are available. If not, does the model rely on external tools (e.g., detectors) to obtain ROI proposals during testing? This is critical for evaluating the practicality of the approach.

---

### **Minor Weaknesses**

1. Notational inconsistencies: e.g., $h_{\text {j}}$uses subscript j, but the first occurrence of $x_{\text {roi}}$ lacks a subscript.
2. Missing t-SNE visualizations of the learned embeddings.
3. Figure 6 lacks comparative visualizations (e.g., against baselines), limiting its interpretability.

**Questions:**

All concerns listed above must be addressed. I must emphasize that the writing quality is severely deficient, particularly in the methodology section. If the authors aspire for this work to meet the standards of a venue like ICLR, a substantial revision focused on clarity, rigor, and completeness of exposition is absolutely necessary.

---

### Official Review · Reviewer_itfz · 2025-11-03

**Soundness:** 1
**Presentation:** 1
**Contribution:** 2
**Rating:** 0
**Confidence:** 4

**Summary:**

The paper addresses two challenges of medical image understanding:
1. the lack of high-quality annotated datasets
2. a method that produces predictions based on small critical details in an image in contrast to global image features.

To tackle the first challenge, the authors use human–AI collaborative annotation to construct a novel dataset called MedRegion-500k (only a subset of which will be publicly released).

For the second challenge, the authors propose RegionMed-CLIP, a region-aware multimodal contrastive learning framework that explicitly incorporates localized pathological signals along with holistic semantic representations.

**Strengths:**

1. Breadth of the dataset, with coverage of various medical imaging modalities and anatomical regions.
2. The model seems to outperform all methods it was compared to.

**Weaknesses:**

1. There is no evidence for the quality and reliability of the resulting annotations provided.
2. The baseline models are fairly old (the latest model is from 2023). It would be advisable to compare the model's effectiveness in comparison to more recent studies both within this domain as well as for generalist VLM models.
3. There is a significant number of inconsistencies in the references (a significant portion of references in the paper refers to non-existing works, including references to the models used as baseline).
4. Negative labels are purely text based. Is there evidence that the model learns to differentiate similar-looking but different diseases from images?
5. It is difficult to disentangle the contribution of the architecture from the dataset without having results of other architectures trained on the same dataset. No proof that the architecture did in fact lead to the demonstrated improvement in performance and not purely the dataset.

**Questions:**

1. What was the size of the fine-tuning train and test datasets? What is the fine-tuned model performance on the test dataset? Were there any additional tests validating the quality of the resulting MedRegion-500k?
2. Can the model process multiple regions? What is the distribution of the number of regions per image in the dataset? What is the average size of the region? What is the distribution of region sizes in the dataset? Does the model perform better or worse when the region is small or big compared to the baseline models?

---

### Note · Authors · 2025-11-12

I have read and agree with the venue's withdrawal policy on behalf of myself and my co-authors.